# Activity Changes of the Peptic Lactoferrin Hydrolysate in Human Gastric Cancer AGS Cells in Response to Cu(II) or Mn(II) Addition

**DOI:** 10.3390/foods12142662

**Published:** 2023-07-11

**Authors:** Li-Ying Bo, Zhi-Qin Pan, Qiang Zhang, Chun-Li Song, Jian Ren, Xin-Huai Zhao

**Affiliations:** 1Faculty of Food Quality and Safety, Qiqihar University, Qiqihar 161006, China; 02540@qqhru.edu.cn (L.-Y.B.); zhiqinpan_123@163.com (Z.-Q.P.); songchunlilily@sina.com (C.-L.S.); 2School of Biology and Food Engineering, Guangdong University of Petrochemical Technology, Maoming 525000, China; zhangqiang@gdupt.edu.cn; 3Maoming Branch, Guangdong Laboratory for Lingnan Modern Agriculture, Guangdong University of Petrochemical Technology, Maoming 525000, China

**Keywords:** lactoferrin hydrolysate, copper, manganese, AGS cells, bioactivity

## Abstract

Lactoferrin is an interesting bioactive protein in milk and can interact with various metal ions of trace elements such as copper, iron, manganese, and others. In this study, a lactoferrin hydrolysate (LFH) was generated from commercial bovine lactoferrin by protease pepsin, fortified with Cu^2+^ (or Mn^2+^) at two levels of 0.64 and 1.28 (or 0.28 and 0.56) mg/g protein, respectively, and then measured for the resultant bioactivity changes in the well-differentiated human gastric cancer AGS cells. The assaying results indicated that the LFH and Cu/Mn-fortified products had long-term anti-proliferation on the cells, while the treated cells showed DNA fragmentation and increased apoptotic cell proportions. Regarding the control cells, the cells treated with the LFH and especially Cu/Mn-fortified LFH had remarkably up-regulated mRNA expression of caspase-3 and Bax by respective 1.21–3.23 and 2.23–2.83 folds, together with down-regulated mRNA expression Bcl-2 by 0.88–0.96 folds. Moreover, Western-blot assaying results also indicated that the cells exposed to the LFH and Cu/Mn-fortified LFH (especially Mn at higher level) for 24 h had an enhanced caspase-3 expression and increased ratio of Bax/Bcl-2. It can thus be concluded that the used Cu/Mn-addition to the LFH may lead to increased bioactivity in the AGS cells; to be more specific, the two metal ions at the used addition levels could endow LFH with a higher ability to cause cell apoptosis by activating caspase-3 and increasing the Bax/Bcl-2 ratio.

## 1. Introduction

One minor component of milk proteins is lactoferrin that is a single polypeptid glycoprotein with a molecular weight of about 80 kDa [1,2]. Native lactoferrin usually chelates with Fe(III) and also has an ability to bind other trace metals such as iron, zinc, copper, manganese, chromium, and aluminum [1,2]. It is well known at present that lactoferrin possesses various bioactivities such as anti-microbial, anti-carcinogenic, anti-inflammatory, and immuno-modulatory effects, which are dependent on the lactoferrin structure of and the bounded metal ions [1,3,4]. From a chemical point of view, proteins have functional groups (e.g., NH, NH_2_, OH, and SH) and thus can interact with trace metal ions, leading to a changed property. Moreover, lactoferrin upon being hydrolyzed by pepsin will yield the important product lactoferricin B, which has been proved to cause cell apoptosis in two gastric cancer cells [5,6]. A previous study of our group also indicated that fortified metal (Cu^2+^ or Mn^2+^) lactoferrin hydrolysates were responsible for apoptosis of AGS and BGC cells [7,8].

Stomach cancer is the most frequently diagnosed malignant tumor around the globe, and has high morbidity and mortality in countries such as Russia, Korea, and Japan. Gastric cancer can be successfully cured if tumors are diagnosed at an early stage and removed surgically [9]. However, gastric cancer is often not found until cancer cells have already spread to other tissues or organs. Outcomes are not very good for cancer patients with advanced gastric cancer that cannot be removed by surgery or chemotherapy. Malignant tumors with distant metastasis are not mostly curable, and chemotherapy, usually by using 5-fluorouracil (5-FU) together with other anti-cancer agents, shows little curative effect and produces unbearable pain for patients. Hence, there is an urgent need to find natural compounds that can potentially treat or prevent gastric cancer.

Several peptides from natural food hydrolysates have aroused some researchers’ great interest; among these assessed proteins is lactoferrin (LF) that exists in milk and dairy foods. Previous studies had proposed that both LF and a LF derivative lactoferricin B had in vitro anti-cancer activities against two gastric cancer SGC-7901 and AGS cells [8,10]; their results also proved that both LF and lactoferricin B had pro-apoptosis to the cells. Recently, researchers have also paid special attention to the potential interaction between proteins and other dietary components. The interactions between proteins/peptides and other substances have been evidenced to impact their bio-functions. Proteins belong to diverse functional groups (e.g., NH, NH_2_, OH, and SH) and thus can interact with some trace metal ions, leading to changed bioactivities [11]. As is known, these trace metal ions themselves have benefits for human health [12]. It has been found that Lactoferrin (LF) supplemented with Fe^3+^ led to greater growth inhibition in the HepG2 cells infected with the hepatitis B virus (HBV) as compared to LF itself, and could cause apoptosis in the cells [13]. However, the study on protein/peptide–metal interactions has still not been efficiently investigated for trace elements such as Mn, and whether this interaction may cause changed activity to cancer cells remains a challenge. It is meaningful to verify how the interaction between these essential trace metal ions and peptic lactoferrin hydrolysate (LFH) leads to changed anti-cancer activity in gastric cancer cells, because LF in the stomach will be digested by pepsin to generate this peptic LFH.

Two microelements, copper and manganese, are accepted as essential nutrients for the body. Cu(II) is a cofactor for several Cu-dependent enzyme systems such as dopamine β-hydroxylase, cytochrome-c, thyroxinase, extracellular superoxide dismutases, cytosolic, lysyloxidase, and electron transport proteins [14]. For humans, this microelement is well known as playing a protective role against bone demineralization, leucopenia, demyelinization of nerve tissues, and arterial fragility, related with the activities of Cu-dependent enzymes. Mn(II) is a component of three metalloenzymes, including the critical antioxidative enzyme in the mitochondria, host of other enzymes [15]. Hence, both Cu and Mn are considered as required minerals of high importance for optimal human health. The nutritional importance of the relationship between Cu/Mn and human health has been investigated; while some organic materials are able to bind one or more of copper and manganese to form complexes, and especially bio-activities of its complexes are thus enhanced, such as oxidant, anti-cancer, and immune effects [16,17]. A previous study from Yu and coauthors reported that copper complexes could be used as potential therapeutic drugs for the treatment of some types of cancers; however, clinically approved use of these complexes were only limited to the Wilson patients [18]. In another study, Mn(II) complexes were measured with higher activities in cancer cells, and especially a Mn(II) complex [(Adpa) Mn(Cl)(H_2_O)] exerted enhanced growth inhibition in the glioma cells by inducing mitochondrial dysfunction [19]. If the digested LF in the stomach interacts with trace metal ions such as Cu^2+^ or Mn^2+^, potential activity change in cancer cells can be an interesting but unsolved issue. Thus, its investigation deserves our consideration.

In a previous study, we assessed the impact of Cu^2+^/Mn^2+^ supplementation of a bovine LFH by pepsin on well-differentiated human gastric cancer AGS cells, in which cell apoptosis was evident to be mediated via an intrinsic pathway and autophagy inhibition [7]. In this study, the Cu/Mn-supplemented LFH was also assessed for its in vitro anti-cancer activities against the AGS cells including the long-term anti-proliferative effect by cell colony assay, apoptotic induction by DNA fragmentation and flow cytometry assays, and especially expression regulation of several apoptosis-related genes and proteins, using LFH as a control. Moreover, several apoptosis-related proteins were verified for their expression changes in the cells treated with a caspase-3 inhibitor and the Cu/Mn-supplemented LFH using a Western-blot assay, to further disclose the anti-cancer mechanism.

## 2. Materials and Methods

### 2.1. Chemicals and Reagents

The bovine LF used for BLH preparation was bought from MILEI Gmbh (Leutkirch, Germany) with corresponding protein and iron contents of 979.0 g and 170 mg per kilogram. The fetal bovine serum (FBS) and Dulbecco’s modified Eagle’s medium (DMEM) used for cell experiments were bought from Wisent Inc. (Montreal, QC, Canada) and Sigma-Aldrich Co., Ltd. (St. Louis, MO, USA). The porcine gastric mucosa pepsin (CAS: 9001-75-6) were also bought from Sigma-Aldrich Co., Ltd. (St. Louis, MO, USA). Dextran T-70, phosphate-buffered saline (PBS), crystal violet dye, and TBST (Tris-buffer saline with 0.1% Tween-20) were all bought from the Beyotime Institute of Biotechnology (Shanghai, China). Two kits, the Rremix Ex TaqTM kit and the PromeScriptTM RT Reagent Kit SYBR^®^ were bought from Takara Bio Ltd. (Kusatsu, Japan), while the RNAprep pure cell kit was provided by the Tiangen Biochemical Technology Co., Ltd. (Beijing, China). Other chemicals were analytical agents. The ultrapure water used in this study was produced with Milli-Q Plus (Millipore Corporation, New York, NY, USA).

### 2.2. Sample Preparation

LFH was prepared as previously described with a minor modification [20]. In brief, 5.0 g LF was dissolved in water, adjusted with 1 mol/L HCl to pH 2.5 and a final protein concentration of 0.05 g/mL, and hydrolyzed with pepsin (750 units/g protein) at 37 °C for 4 h with constant stirring. After hydrolysis, the yielded LF hydrolysate was heated to 80 °C, held at this temperature for 15 min to inactivate the pepsin, cooled to 22 °C, adjusted with 1 mol/L NaHCO_3_ to pH 7.0, and centrifuged (12,000× *g*) at 4 °C for 30 min, followed by a freeze-drying treatment using an ALPHA 1-4 LSC plus freeze dryer (Marin Christ, Osterode, Germany). The freeze-dried LFH was ground into powder, and kept at 20 °C before use.

Copper- and magnesium-supplemented LFH was prepared by LFH with different concentrations of CuCl_2_ (or MnSO_4_) at 0.64–1.28 (or 0.28–0.56) mg/g protein, according to the previous method with minor modifications [21]. These mixtures were adjusted with 1 mol/L NaHCO_3_ to pH 7.0, held at 22 °C for 1 h, followed by dialysis with deionized water to remove extra salt, and then freeze-dried, ground into powder, and also kept at 20 °C until use. In chemistry, these used Cu/Mn-supplementation levels were equivalent to Cu 10–20 and Mn 5–10 µmol/g protein, respectively. In this study, Cu-supplementation of 10 and 20 µmol/g protein of LFH was designated to yield Mixture I and Mixture II, while Mn-supplementation of 5 and 10 µmol/g protein of LFH was designated to generate Mixture III and Mixture IV, respectively.

### 2.3. Cell Culture Conditions

The AGS cells were obtained from the Cell Bank of the Chinese Academy of Sciences (Shanghai, China). As a cell provider requires, AGS cells were cultured in the DMEM medium supplemented with penicillin of 100 units/mL, streptomycin of 100 g/mL, and FBS of 10%. The medium was replaced every day. The cells were suggested to be cultured at 37 °C in a humidified incubator with 5% CO_2_.

### 2.4. Colony Formation Assay

The cells (1 × 10^3^ cells/well) were seeded in the 6-well plates to assess the long-term anti-proliferative activities of the prepared LFH and Mixtures I–IV. The cells exposed to the medium consisting of these samples (25 mg/mL dose level) were cultured for 24 h, and the medium consisting of 5% FBS was replaced every three days. The cells were cultured for 10 or 20 d. Subsequently, the medium was removed, and methanol was added into the wells to fix the cells. After a staining with crystal violet solution, the cells were dried overnight and photographed with a digital camera (Type EOS 6D, Canon, Tokyo, Japan).

### 2.5. Assay of DNA Fragmentation by Gel Electrophoresis

The cells (1 × 10^6^ cells per well) seeded in the 6-well plates with 2 mL medium were cultured for 24 h. The cells were then exposed to 2 mL/well fresh medium including these samples (25 mg/mL) for 24 h. The treated cells were then subjected to DNA extraction, according to the DNA extraction kit’s instructions. The cells treated with the medium containing 5% FBS only were used as a negative control. The cells were harvested using the trypsin-EDTA, washed with the 10 mmol/L PBS (pH 7.3) twice, and lysed using the DNA lysis buffer (5 μL proteinase K and 1 mL RNase A per 106 cells) at 55 °C for 12 h. An equal volume of phenol was then used to extract DNA, using a 12,000× *g* centrifugation at 4 °C for 20 min. This extracting treatment was repeated twice. Chloroform of an equal volume was then added to the obtained upper liquid, thoroughly mixed, and then centrifuged at 12,000× *g* and 4 °C for 20 min. Subsequently, 60 μL of ammonium acetate was added into the supernatant of 300 μL. The supernatant was precipitated by 600 μL ethanol, and kept at 4 °C overnight. The obtained extracts were centrifuged at 4 °C for 10 min and shaken for 15 s; a small amount of DNA precipitation then appeared. After centrifuging the extracts again and removing the supernatant, the precipitates were dispersed in the Tris-EDTA buffer of 20 μL and separated by using 1.5% agarose gel electrophoresis to reveal DNA fragmentation.

### 2.6. Assay of Apoptotic Induction

The cells (2 × 10^4^ cells/well) seeded in 6-well plates with 2 mL medium were incubated for 24 h. After medium discarding, the cells were exposed to the medium containing these samples (25 mg/mL) or 5% FBS (as negative control) for 24 h. After these treatments, an AnnexinV-FITC/PI apoptosis detection kit was used following the kit’s instructions. The harvested cells were resuspended in 0.5 mL of the annexin V-FITC binding buffer containing 5 μL annexin V-FITC and 10 μL PI at 20 °C for 30 min in the dark, and detected by the flow cytometry (FACS Calibur, Becton Dickson) to obtain the intact (Q3), early apoptotic (Q4), late apoptotic (Q2), and necrotic (Q1) cell proportions.

### 2.7. RT-PCR Assay

The cells were cultured with 2 mL medium containing these samples (25 mg/mL) for 24 h in the 6-well plates. The total RNA was extracted from the treated cells using the RNAprep pure cell kit (Tiangen Biochemical Technology Co., Beijing, China) following the kit manufacturer’s guidelines. Total RNA was then inversely transcripted into the complementary DNA (cDNA) to ensure RNA purity, during which 1 ng total RNA and the PrimeScriptTM RT reagent kit (Takara Bio Ltd., Kusatsu, Japa) were used for cDNA synthesis. A real-time PCR assay was carried out on a real-time PCR thermal cycler system (Life Technologies Corporation, Carlsbad, CA, USA), using the SYBER Green qPCR Kit (Takara, China) and following the kit manufacturer’s instructions. The mixtures went through 40 cycles of amplification for 30 s at 95 °C, followed by 30 s at 60 °C. The primer sequences used in this study are given in Table 1. The relative expression levels of RNA were calculated based on the 2^−∆∆Ct^ method, while the β-actin expression was used as a control to normalize each sample [22]. All primers were designed by the Sangon Biotech Company (Shanghai, China).

### 2.8. Western-Blot Assay

The cells (5 × 10^6^ cells/dish) were cultured on 100-mm cell culture dishes with 10 mL medium containing 5% FBS at 37 °C for 24 h, treated by the medium containing these samples (25 mg/mL) for another 24 h, washed twice in the cold PBS, and then lysed on ice for 30 min with 100 μL RIPA buffer containing PMSF of 1 mmol/L. The lysate was centrifuged at 12,000× *g* and 4 °C for 5 min. The resultant supernatant was served as a total cellular protein and the protein concentration was detected with the BCA protein assay kit according to the kit’s instructions. Separation of equal-amount proteins was achieved by using a 10–15% SDS-PAGE gel. Subsequently, the proteins were shifted to the PVDF membrane. The membrane was blocked with 5% BSA for 2 h, and incubated with the primary anti-body (1:3000 dilution) in blocking buffer overnight at 4 °C. The membrane was then washed three times with the TBST, incubated with the corresponding secondary anti-body (1:3000 dilution) at 20 °C for 2 h, and finally washed three times with the TBST. The immunoreactive intensity values of the bands were measured using the Chemi Scope 6300 (Clinx Science Instrument, Shanghai, China).

### 2.9. Statistical Analysis

All data were collected from three separate experiments and assays. Statistical analysis was performed by the SPSS software (SPSS Inc., Chicago, IL, USA) using Duncan’s multiple range tests.

## 3. Results

### 3.1. Effects of LFH and Mixtures I–IV on Colony Formation of AGS Cells

In the previous study, proliferative inhibition of LFH and Mixtures I–IV in the cells were assessed [8]. Thus, these data were not reported in the present study. For the following other evaluation indices, the dose level of all assessed samples for cell treatment was fixed at 25 mg/mL while the cell treatment time was selected at 24 h, based on the results of the previous results. When the cells were exposed to these samples to compare their long-term anti-proliferative activities (10 or 20 d), the results (Figure 1) showed that Mixtures I–IV also had more obvious anti-proliferative effects against the cells than the LFH alone. In comparison with the cells treated with Mixtures I–II, cell colony numbers and sizes of the cells exposed to Mixtures III–IV were significantly reduced, while the cells treated with Mixture IV (or Mixture II) had fewer cell colony numbers and smaller sizes than those treated with Mixture III (or Mixture I). That is, Mn was more active than Cu to elevate long-term anti-proliferative effect in the cells, and higher metal supplementation level caused stronger long-term growth inhibition on the cells.

### 3.2. DNA Fragmentation and Cell Apoptosis in Response to LFH and Mixtures I–IV

To explore whether these samples had potential apoptotic induction to the cells, a DNA fragmentation assay was performed. Cell treatment with LFH and Mixtures I–IV (25 mg/mL) for 24 h resulted in clear DNA fragmentation, in comparison with the control cells (Figure 2). In total, it was seen from the assay results of DNA fragmentation that the LFH had weaker apoptotic induction than Mixtures I–IV (line 2 versus lines 3–6), and Mixtures III–IV may induce more obvious apoptosis in the cells than Mixtures I–II (lines 5 and 6 versus lines 3 and 4). Thus, further evaluation was needed to verify the apoptotic induction of these samples, for example, using the flow cytometry technique together with the Annexin V-FITC and PI double staining to measure the proportions of the intact, early apoptotic, late apoptotic, and necrotic cells.

The measured total apoptotic cell (Q2 + Q4) proportions were used in the present assay to reflect the apoptotic induction of these samples. Flow cytometry analysis results (Figure 3) showed that all assessed samples were able to induce cell apoptosis. Compared with control cells (total apoptotic cell proportion 15.8%), the cells exposed to the LFH for 24 h had an enhanced total apoptotic proportion of 35.9%. Moreover, the cells exposed to Mixtures I–II had total apoptotic proportions of 40.8–49.9%, while those exposed to Mixtures III–IV had total apoptotic proportions of 58.4–70.7%. Clearly, Mixtures III–IV showed a higher apoptotic induction than Mixtures I–II.

### 3.3. Effects of LFH and Mixtures I–IV on the Expression of Apoptosis-Related Genes

To elaborate the possible molecular mechanism involved in the bioactivity of these samples in the cells, expression levels of several genes related to cell apoptosis were examined. The results (Table 2) showed that all samples could up- or down-regulate mRNA expression of these genes. Compared with control cells, the LFH-treated cells had up-regulated caspase-3 (1.21-fold) and Bax (2.23-fold) but down-regulated Bcl-2 (0.96-fold). The cells exposed to Mixture I showed up-regulated caspase-3 (1.65-fold) and Bax (2.35-fold) but down-regulated Bcl-2 (0.95-fold), while those exposed to Mixture II showed much up-regulated caspase-3 (2.34-fold) and Bax (2.54-fold) but highly down-regulated Bcl-2 (0.92-fold). That is, Mixture I and especially Mixture II were more effective than the LFH to regulate the expression of the three genes. At the same time, the cells exposed to Mixture III had 3.07- and 2.64-fold increases in caspase-3 and Bax, but 0.90-fold decreases in Bcl-2, while those exposed to Mixture IV showed 3.26- and 2.83-fold increases in caspase-3 and Bax, but 0.88-fold decreases in Bcl-2. Mixture III and especially Mixture IV thus had greater potential than Mixtures III to regulate the three genes.

Mixtures I–II, LFH fortified with Cu^2+^ 0.64 or 1.28 mg/g protein; Mixtures III–IV, LFH fortified with Mn^2+^ 0.28 or 0.56 mg/g protein, respectively. Gene expression is normalized against β-actin.

### 3.4. Bax, Bcl-2, and Caspase-3 Protein Expression in Response to LFH and Mixtures I–IV

Expression changes of three proteins (Bax, Bcl-2, and caspase-3) in the cells exposed to these samples were also detected. After cells treatment for 24 h, caspase-3 and Bax expression levels were significantly up-regulated, while the Bcl-2 expression level was markedly down-regulated (Figure 4A). The ratio of Bax/Bcl-2 was also remarkably increased (Figure 4B). Moreover, Mn supplementation showed greater ability than Cu supplementation to regulate the three proteins. The results also indicate that a higher Cu/Mn supplementation level resulted in more remarkable regulation on the three proteins. These results indicated that apoptotic induction of the LFH or Mixtures I–IV may come from the mitochondrion-mediated regulation and caspase-3-dependent apoptosis pathway.

To further explore whether apoptosis of the cells was triggered by the hypothetic pathway, the cells were co-cultured in the medium containing a caspase-3 inhibitor (z-VAD-fmk) and Mixture II or Mixture IV. The expression level of Bax (or Bcl-2) in the Mixture II- or Mixture IV-treated cells was higher (or lower) than that in the z-VAD-fmk-treated cells alone, and the expression level of caspase-3 was significantly decreased (Figure 5). These results revealed that Mixture II and Mixture IV could promote cell apoptosis through the caspase-3-dependent and mitochondrion-mediated pathway, which was consistent with the assay results from the real-time RT-PCR.

## 4. Discussion

The findings from the present study verified in vitro anti-cancer activities of LFH and Cu/Mn-fortified LFH in human gastric cancer AGS cells. The anti-proliferative effect of the LFH and the prepared Mixtures I-IV was evaluated by a cell colony formation assay in vitro, and Cu/Mn supplementation was evident to strengthen LFH’s anti-proliferative effect on AGS cells. Consistent with the present results, a pure single peptide from soy beans was observed with an inhibitory effect on both human blood and colon cancer cells, and the protein hydrolysates from Sri Lankan rice brans were also found to have anti-proliferation and cytotoxicity to human lung cancer NCI-H460 and cervical cancer HeLa cells [23,24]. In another study of our group, Cu/Mn-fortified LFH was also verified with higher anti-proliferative activities against low-differentiated human gastric cancer BGC-823 cells.

Apoptosis is a homeostatic mechanism to balance cell death and cell division and to preserve appropriate cell numbers in the body, and is a typical process of programmed cell death occurring in the multi-cellular organisms [25,26]. The character of apoptosis includes nuclear fragmentation, cell shrinkage, chromosomal DNA fragmentation, and chromatin condensation [27]. Several peptides (hydrolysates) derived from food protein, such as the LFcinB25, the peptides from donkey milk hydrolysate, the peptides from algae protein waste, and protein hydrolysate of a giant grouper, all have obvious anti-cancer activity and can effectively promote apoptosis in various cancer cells [28,29,30,31]. In a reported study, lactoferricin showed ability to cause DNA fragmentation of human B-lymphoma cells and induce cell apoptosis [32]. There is a report that says that lactoferricin was evident in promoting the apoptosis of colon cancer cells [33]. It was also found that the hydrolysates from roe could promote the apoptosis of human oral cancer Ca9-22 and CAL27 cells [31]. In this study, flow cytometry assay results using annexin V and PI double-staining demonstrated that LFH and Cu/Mn-endowed LFH both caused the apoptosis of AGS cells while the treated cells also displayed typical apoptotic morphology. Moreover, Cu/Mn conferred the resultant mixtures with a stronger ability to induce AGS cells apoptosis, especially using Mn supplementation and a high metal supplementation level. This fact indicates that some peptide fragments derived from LFH may interact with the metal ions Cu^2+^/Mn^2+^ to generate corresponding peptide–metal complexes, thereby producing higher activities in the cells.

Further evaluations focused on molecular mechanisms of AGS cells apoptosis. Apoptosis is a necessary process of cell death and, therefore, apoptotic induction of tumor cells is an effective strategy for cancer treatment [34,35]. Although cell apoptosis is involved in many pathways for caspases activation, only two pathways have been studied in detail [36]; that is, the extrinsic (death receptor) and intrinsic (mitochondrial) pathways. Death receptors, through adaptor molecules, recruit initiator caspases, pro-caspases cleave, and then activated caspases are able to connect ligand binding on the cell surface, resulting in apoptotic induction [37,38]. The intrinsic pathway refers to the participation of mitochondrial, intrinsic death signals that stimulate mitochondria, which release activated caspase proteins into the cytosol, activate caspase-9, and finally trigger cell apoptosis [39]. Initiator caspases cleave caspases and then activate effector caspases (such as caspase-3, caspase-6, and caspase-7) [40]. In this study, LFH and Mixtures I–IV induced typical DNA fragmentation and apoptosis in AGS cells, which seemed to be associated with the mitochondria-mediated apoptosis pathway via caspase-3 activation. Moreover, Mn supplementation and a high metal supplementation level brought stronger apoptotic induction, compared to Cu supplementation and a low metal supplementation level. This finding may indicate that the molecular mechanism of the evaluated apoptotic induction seems to be related to the intrinsic apoptotic pathway. Thus, further research needs to be performed to explore the molecular mechanism, which may make a contribution to the apoptosis of AGS cells.

The most important regulatory proteins of the mitochondrion-mediated apoptotic pathway are those from the Bcl-2 protein family [41]. The members of the Bcl-2 protein family include Bcl-2, Bfl-1, Bcl-XL, Bcl-W, Mcl-1, and Bfl-1 possessing an anti-apoptosis character, together with Bad, Bax, Bcl-Xs, Bak, Bid, Bim, Hrk, and Bik possessing a pro-apoptosis character. However, the apoptotic induction is more relevant to the balance of Bax and Bcl-2, compared with Bcl-2 quantity alone [42]. In general, the ratio of Bax and Bcl-2 serves as an index of cell apoptosis [43]. A previous study verified that sepia ink oligopepide (a tripeptide) induced apoptosis of prostate cells by elevating Bax/Bcl-2 ratio [44]. Another peptide derived from sweet potato could activate caspase-3, down-regulate Bcl-2 expression but up-regulate Bax expression in colon cancer HT-29 cells [45]. Moreover, it also has been successfully demonstrated that a peptide from soy hydrolysate caused apoptosis of colon cancer HT-29 cells by increasing the Bax/Bcl-2 ratio and activating caspase-3 [46]. In this study, if the cells were exposed to Cu- and especially Mn-supplemented LFH, the pro-apoptotic protein Bax was remarkably up-regulated while the anti-apoptotic protein Bcl-2 was significantly down-regulated, resulting in an increased Bax/Bcl-2 ratio (Figure 4B). This phenomenon was consistent with those in the mentioned studies.

## 5. Conclusions

Bovine lactoferrin hydrolysate and its Cu/Mn-fortified products had enhanced in vitro anti-cancer activity in human gastric cancer AGS cells, especially when Mn supplementation and a high metal supplementation level were used. These supplemented lactoferrin hydrolysates showed a stronger anti-proliferative effect on the cells and caused greater cell apoptosis. Moreover, the Mn-supplemented hydrolysate and high metal supplementation level could remarkably up- and down-regulate the apoptosis-related proteins Bax and Bcl-2. Thus, this study verifies a potential anti-cancer benefit of the two trace metal ions on bovine lactoferrin, and gives an insight into the interaction between LF/LFH and trace metals.

## Figures and Tables

**Figure 1 foods-12-02662-f001:**
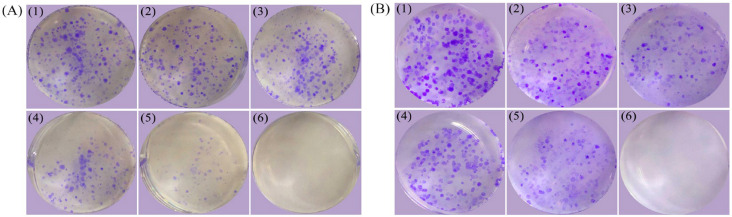
Colony formation assay of the control cells (1) and the cells exposed to the LFH (2) or Mixtures I–IV (3–6) with culture times of 10 (**A**) or 20 (**B**) days.

**Figure 2 foods-12-02662-f002:**
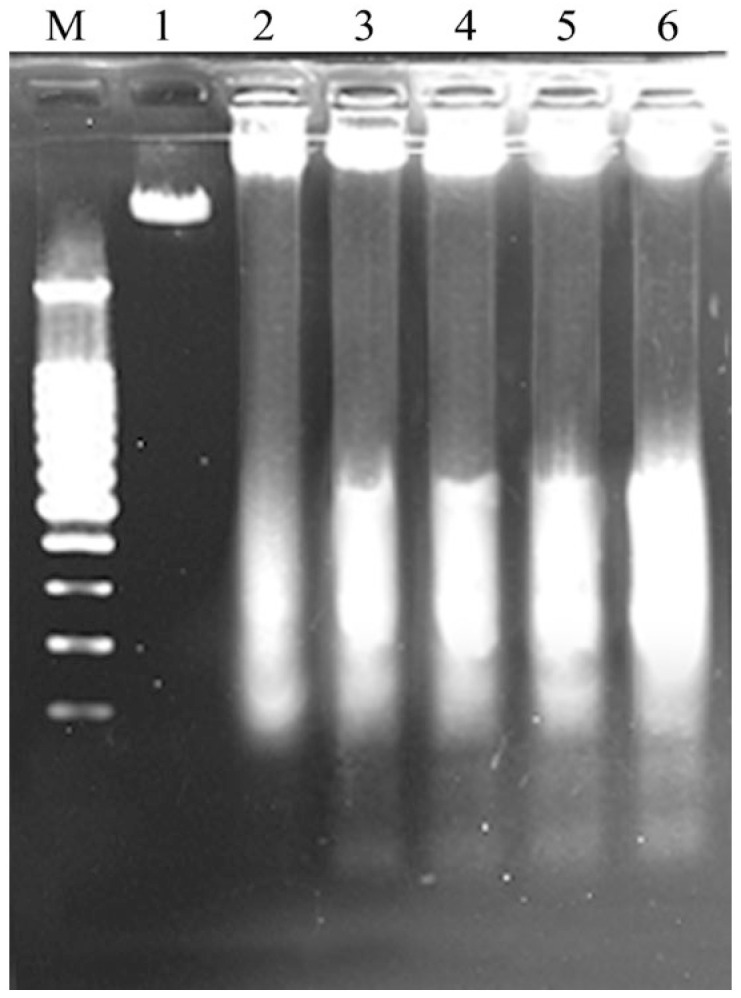
DNA fragmentation of the control cells (lane 1), the cells treated with the LFH (lane 2) and Mixtures I–IV (lanes 3–6), respectively. Lane M, DNA markers.

**Figure 3 foods-12-02662-f003:**
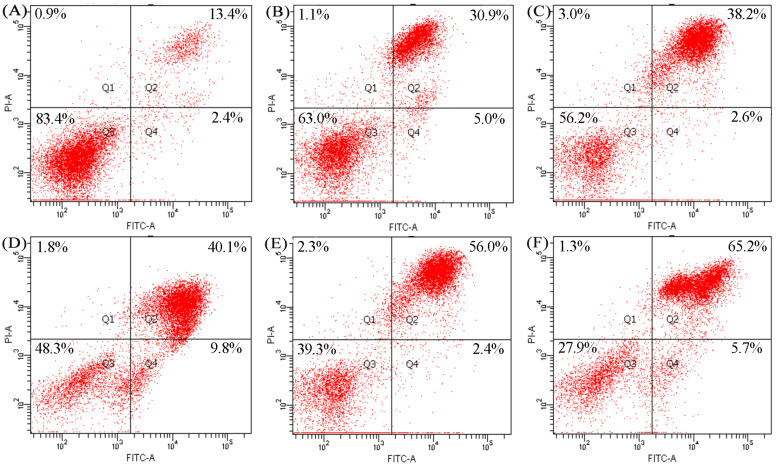
Apoptotic induction of the LFH and Mixtures I–IV to the cells. The cells were without any treatment (**A**), or treated with LFH (**B**) or Mixtures I–IV (**C**–**F**) at 25 mg/mL for 24 h.

**Figure 4 foods-12-02662-f004:**
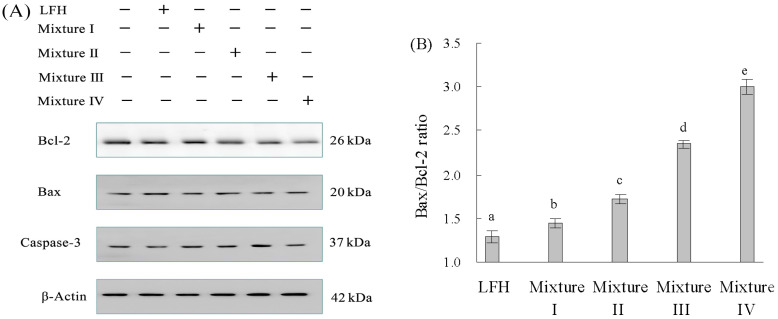
Expression changes of three apoptosis-associated proteins in the treated cells (**A**) and calculated ratio of Bax/Bcl-2 expression (**B**). The different letters such as a, b, c, d, and e above the columns show that the means of different groups differ significantly (*p* < 0.05) by using one-way analysis of variance.

**Figure 5 foods-12-02662-f005:**
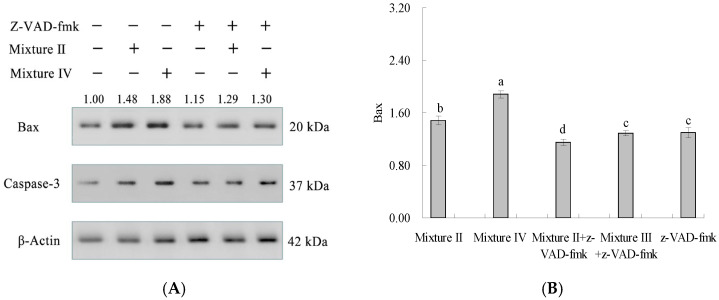
Effects of the Mixture II, Mixture IV, and the caspase-3 inhibitor z-VAD-fmk on the expression of three apoptosis-related proteins in the cells (**A**) and calculated analysis of Bax expression level (**B**). The different letters such as a, b, c, and d above the columns show that the means of different groups differ significantly (*p* < 0.05).

**Table 1 foods-12-02662-t001:** The primers used in RT-PCR assay.

Genes	Primer Sequences (5′-3′)	Primer Lengths (bp)
Bax	Forward GAAGACCTGGATGCTCTGGAACAC	159
	Reverse GAGTACCACGAAGGCACAACTGAC	
Bcl-2	Forward ACATCCATTATAAGCTGTGGCAGAGG	166
	Reverse TGCAGCGGCGAGGTCCTG	
Caspase-3	Forward TTGAGACAGACAGTGGTGTTGATGATG	98
	Reverse ATAATAACCAGGTGCTGTGGAGTATGC	
β-Actin	Forward GGAGCTGCCGTTATACTGTTCTGG	240
	Reverse TGCCTCCTGTGTCTTCAATCTTGC	

**Table 2 foods-12-02662-t002:** Detected expression changes of the three apoptosis-related genes in the cells.

Genes	LFH	Mixture I	Mixture II	Mixture III	Mixture IV
Bax	2.23 ± 0.07	2.35 ± 0.05	2.54 ± 0.05	2.64 ± 0.05	2.83 ± 0.03
Bcl-2	0.96 ± 0.03	0.95 ± 0.04	0.92 ± 0.02	0.90 ± 0.03	0.88 ± 0.07
Caspase-3	1.21 ± 0.10	1.65 ± 0.18	2.34 ± 0.08	3.07 ± 0.05	3.26 ± 0.01

## Data Availability

The data presented in this study are available on request from the corresponding author.

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
