# Peer review of "Activity Changes of the Peptic Lactoferrin Hydrolysate in Human Gastric Cancer AGS Cells in Response to Cu(II) or Mn(II) Addition"

_foods, 2023, doi:10.3390/foods12142662_

Round 1
Reviewer 1 Report
The authors evaluated that the effects of lactoferrin hydrolysate (LFH) and Cu/Mn-fortified LFH on cell growth, apoptosis, and gene and protein expressions related to the apoptosis in the well-differentiated human gastric cancer AGS cells. These supplemented LFH showed stronger anti-proliferative effect on the cells and caused greater cell apoptosis. Moreover, the Mn-supplemented LFH could remarkably up- and down-regulate the apoptosis-related proteins Bax and Bcl-2. This is an important contribution to the field of anti-cancer and protein function. I have comments, explained below. I hope that my comments are very useful for the improvement of this research.
Major comments
(1) Cu/Mn-fortified LFH: A single metal addition group (Cu and Mn) was not established, and when considering the effect of Cu/Mn-fortified LFH, it is not clear whether the effect of LFH or the effect of the metal alone or synergistically. Therefore, I think that the authors should conduct to add the metal-alone addition groups.
(2) Cu and Mn: Why did you decide to enhance Cu and Mn in LFH? Please indicate your reasons in the text.
(3) Section 2.2: Are copper and manganese present bound to LFH? Or are they separated from each other? Please rewrite the text to clarify, as it is not clear from reading the Methods section. In addition, is there any removal of salt formed during neutralization?
(4) Molecular weight: The molecular weight for LFH is necessary to discuss the results of this experiment. It is desirable to obtain the data of molecular weight.
(5) Statistical analysis: The authors used the Duncan’s multiple range test as statistical analysis. But, Duncan's multiple range test has been pointed out to have problems such as not taking multiplicity. Thus, please change to another multiple tests. Duncan’s multiple range test is very prone to significant differences, so using the correct method should result in no significant differences in many items. Therefore, it is necessary to use the correct statistical analysis. If the statistical processing method changes, the conclusion will also change.
(6) Fig. 4: It is stated that group comparisons are made by ANOVA. This is incorrect; group comparisons cannot be made by ANOVA.
(7) Fig. 5: Like a Fig. 4, the analysis of band intensity is need.
Minor comments
(8) L19-20: Since the values are too detailed, I suggest that they be changed to approximate values.
(9) L32: It would be preferable to indicate specifically what kind of bioactivity it is.
(10)L36: The 2 in “NH2” is a subscript. Other parts need to be checked.
(11)L39: Reference is needed.
(12)L61: Abbreviations in this manuscript are wrong usage; therefore, it should be rechecked throughout. For example, HBV and BLH are an abbreviation used without indicating the official name.
(13)L77-78: I'm curious about the word "excessive". It is inappropriate to call the large number of studies excessive.
(14)L106: igma -> sigma.
(15)L126: Please check the unit of mol/ g protein. The atomic weight of copper is 63.5, so there is approximately 63,500 g of copper in 1 g of protein.
(16)Figure 1: It is difficult to understand the correspondence between the photo and the group.
Author Response
please find it in the attachment

Reviewer 2 Report
The work "Activity changes of the peptic lactoferrin hydrolysate in human gastric cancer AGS cells in response to Cu(II) or Mn(II) addition" presented a very interesting result. However, the author need to reconsider the study design.
1. Characteristics of peptic lactoferrin hydrolysate are required to evaluate. Otherwise, there is no clue to interpret the data whether the activity is from metal or peptide.
2. The mechanism for this change might be the size of peptide which allows molecule can penetrate into cells. That means the fraction of peptide is so important to study.
3. The most critical point is the substance concentration, 25 mg/mL is very high and in the context of un-defined molecule weight of each peptide, it is not reasonable to compare them since the number of molecules are different and the conformation of those are different, too.
Author Response
please find the reply in the word file

Round 2
Reviewer 1 Report
I am satisfied with the revisions that have been made by the authors.
Reviewer 2 Report
Considering this subtance as a food ingridient, this work is acceptable.